# aTBP: A versatile tool for fish genotyping

**Silvia Gianì**[1☯], **Silvia Silletti**[1☯], **Floriana Gavazzi**[1], **Laura Morello**[1], **Giacomo Spinsanti**[2], **Katia Parati**[3], **Diego Breviario**[1]*

**1** Department Scienze Bioagroalimentari, Istituto Biologia e Biotecnologia Agraria, National Research Council, Milano, Italy, **2** Department of Life Sciences, University of Siena, Siena, Italy, **3** Istituto Sperimentale Italiano Lazzaro Spallanzani, Rivolta d'Adda (CR), Italy

☯ These authors contributed equally to this work.
* diego.breviario@ibba.cnr.it

## Abstract

Animal Tubulin-Based-Polymorphism (aTBP), an intron length polymorphism method recently developed for vertebrate genotyping, has been successfully applied to the identification of several fish species. Here, we report data that demonstrate the ability of the aTBP method to assign a specific profile to fish species, each characterized by the presence of commonly shared amplicons together with additional intraspecific polymorphisms. Within each aTBP profile, some fragments are also recognized that can be attributed to taxonomic ranks higher than species, e.g. genus and family. Versatility of application across different taxonomic ranks combined with the presence of a significant number of DNA polymorphisms, makes the aTBP method an additional and useful tool for fish genotyping, suitable for different purposes such as species authentication, parental recognition and detection of allele variations in response to environmental changes.

## Introduction

With approximately 35.000 described species, fishes account for about 50% of all vertebrates. Fish exhibit a great level of diversity, reflecting processes of adaptation to very different aquatic environments. High species number, significant morphological and genetic diversity and environmental fitness, are at the basis of several important scientific issues. These may refer to taxonomy and correct species identification, evolutionary biology and assessment of variation and changes in allele frequencies, resilience and adaptability to extremely variable climate conditions, diversification and parental recognition, traceability of seafood. All these issues find in cellular DNA a common and effective target for investigation. In fact, cellular DNA can potentially be retrieved from any species and any kind of organic substrate, such as muscle, fin, or blood and DNA-based analyses can be applied to any of the issues just mentioned. Species identification is nowadays largely based on DNA barcoding, through the amplification and sequencing of some mitochondrial genes where a sufficient interspecies variation can be detected [1–3]. The fish section of the consortium for barcoding of life (http://www.boldsystems.org/ or https://ibol.org/) includes about 8.000 fish species and relies on the sequence of the 650 bp region of the mitochondrial gene *cytochrome c oxidase I* (*COI*). It represents an effective and comprehensive resource for the analysis of fishes and fish products

**Data Availability Statement:** All relevant data are within the manuscript and its Supporting Information files.

**Funding:** This work was partially supported by the FHfFC (Future Home for Future Communications)

project funded by Regione Lombardia. DB was the recipient. GA: FHfFC 2016 There was no additional external funding received for this study and the funders had no role in study design, data collection and analysis, decision to publish, or preparation of the manuscript.

**Competing interests:** The authors have declared that no competing interests exist.

[4, 5]. More recently, and for the purpose of tracing species in food matrixes that contain a low quality DNA, due to harsh food processing, the use of minibarcodes (shorter fragments of the full length DNA barcode approximately 200 bp long) has been applied with some success. Several minibarcode regions have been identified that allow for differentiation of a range of species, and these regions have also been tested *in silico* to differentiate commercially important salmon and trout species [6–8]. However, limits in the classical DNA barcoding approach may be encountered in the analysis of mixtures composed of multiple species, in the recognition of undeclared substitutions, especially with local varieties, in the availability of specific, known target sequences, and in the need for sequencing and related costs for data elaboration and instrument maintenance.

Genomic DNA data are also very important for conservation management of genetic resources and for assessment of variations occurring in natural populations. This data provides a novel opportunity to investigate how populations have responded to changes, to identify mechanisms underlying these changes, and to evaluate the adaptive potential and vulnerability of populations in the future. A recent and worrisome example has been reported concerning a 60% decline in the populations of salmon of North America and Europe, clearly associated to warmer winter temperatures. Using single nucleotide polymorphisms (SNPs) as molecular tools, declining and near to decline populations have been identified [9]. These declining fish numbers are not only problematic for biodiversity, but their loss also represents an impediment to improving our scientific understanding of key fundamental adaptation strategies revealing molecular responses to life in cold conditions. Cited in the line of our present contribution, this is reminiscent of a well known and early reported adaptation process that explained the occurrence of microtubule polymerization at cold temperatures as dependent on specific amino acid substitutions found in the α- and β-tubulin moieties [10, 11]. In more general terms, the availability of a key functional marker is of importance to monitor the effect of climate changes on population fitness. In this way genomic screening can effectively assess population vulnerability. This has been successfully applied for salmon in a Canadian alpine environment where the maintenance of an almost balanced population of red and white Chinook salmon (*Onchorhynchus tshawytscha*) has been associated to increased carotenoids synthesis and increased heterozygosity at the *major histocompatibility complex* loci [12, 13], respectively. In addition, the reproduction system can obviously affect variation in natural populations and thus the use of suitable molecular markers like polymorphic microsatellite loci and *COI* can help in assigning parentage, in identifying hybridization events and in recording the breeding system [14, 15].

As previously reported, different molecular markers may be utilized for different purposes. Thus, we want to direct the attention to a relatively new molecular marker, animal Tubulin Based Polymorphism (aTBP; [16]), sufficiently versatile to assist these many different purposes. Based on the natural occurrence of polymorphisms in the intron length and nucleotide composition of the β-tubulin genes, the approach may offer an attractive and workable alternative to the genetic identification of fish species, as well as subpopulations and local varieties, with no need for sequencing. Hereby, we present experimental evidence in favour of the use of aTBP for fish genotyping and discuss its possible applications.

## Materials and methods

### Experimental samples

Total DNA extracts made from the following fish species: *Sparus aurata*, *Dicentrarchus labrax*, *Oncorhynchus mykiss*, *Acipenser naccarii*, *Thunnus thynnus*, *Salmo carpio* and *Salmo trutta* f. *fario* were provided by the Spallanzani Institute (Rivolta d'Adda, Italy). These were obtained

from different research projects in which the Institute has been involved: Competus—CRAFT-017633; Cobice—LIFE–04NAT/IT/000126; FP7-SME-2010-1-262523; FP4-FAIR989211; Salvacarpio–Regional project n. 1220; MIIPAF, Three-year plans for fishing and aquaculture—VI 2000–02. The DNA extracts were originally produced from fin-clipped samples by using the semi-automatic BioSprint 96 DNA system (QIAGEN) and the BioSprint 96 DNA Blood Kit (Qiagen) following the manufacturer's protocols. Fish species identification of these samples was performed by the use of a panel of Single Sequence Repeats markers (SSRs), as reported [17–21], with the exception of *T. thynnus* and *O. mykiss*. 15–20 samples of each species were randomly chosen and used for the aTBP molecular analyses. The DNA samples identified by the prefix FT were instead provided by the Life Sciences Department of the University of Siena. These included 34 fish specimens, purchased frozen from local Tuscan markets, consisting of 6 specimens of *Sparus aurata*, 2 of *Acipenser transmontanus*, 6 of *Thunnus albacares*, 4 of *Pangasisus hypophthalmus*, 8 of *Salmo salar*, 4 of *Oncorhynchus mykiss*, and *4 of Dicentrarchus labrax*. Total DNA extractions were performed by using the Wizard® SV Genomic DNA Purification System (Promega), following the manufacturer's instruction for animal tissues. Fish species identification of the FT samples was obtained by DNA sequencing of the fragments amplified with the use of the following universal primers: 5'-TCAACYAATCAYA AAGATATYGGCAC- 3' for the forward and 5'-ACTTCYGGGTGRCCRAARAATCA-3' for the reverse, known to target a conserved portion of the *COI* gene [1–3]; DNA sequencing was performed on both strands and sequences matched to each other. Unaligned and aligned *COI* sequences are provided in the S1 Data.

## aTBP amplification and capillary electrophoresis

30 ng of any total DNA sample, previously characterized either by SSRs or *COI*, were used as template for aTBP PCR amplification. PCR conditions and primer sequences for amplification of intron III (aFex3.2 and aRex3.2) have been recently reported [16]. The forward primer was labeled in 5' position as described in [22]. Two negative controls (no template) were always included in each PCR reaction and all PCR amplifications were repeated at least twice to check the consistency of the amplification profile. 4 μl of each PCR reaction was preliminary loaded on a 2% agarose gel, stained by Atlas Clear Sight DNA Stain (1μg mL$^{-1}$) (Bioatlas) and compared to gene Ruler™1 Kb plus ladder as reference, to verify the intensity of the amplification signal to proceed with the appropriate dilutions to be used for amplicon resolution analysis done by capillary electrophoresis. 2μL of each diluted sample was mixed with 0.2 μl of 1200 LIZ Size Standard and 17.8 μl Hi-Di formamide to a final volume of 20 μL. Samples were denaturated at 95°C for 5 min and, after cooling to -20°C, were loaded onto the ABI 3500 Genetic Analyzer (Thermo Fisher Scientific) for CE separation following the running protocol described by [23].

## Data analysis

The amplicons resolution data were collected using the Data Collection Software v. 3.1 (Thermo Fisher Scientific) and then analyzed by the Gene Mapper Software v. 5.0 tool (Thermo Fisher Scientific). Data analysis was made by comparison of the numerical output of the ABI 3500 analyzer, converted in an excel spreadsheet which allows the association of each specific amplicon profile to each fish species. At least two different electrophoretic runs were performed for each amplified product in order to confirm the aTBP profile. The PCA analysis was carried out with Past3 software [24] based on a presence-absence matrix, obtained from the score of the aTBP markers.

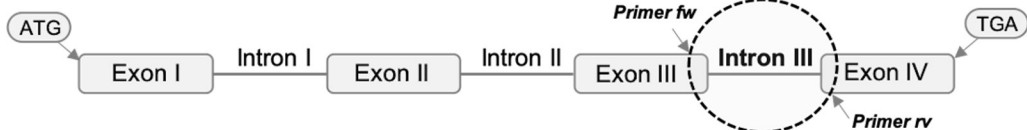

**Fig 1. aTBP: An intron length polymorphism—Based method.** The genomic organization of a generic vertebrate β-tubulin gene is shown. Intron III is used as the source of DNA polymorphism in a PCR reaction that is triggered by the use of the aTBP primer pair.

## Results

As shown in Fig 1, the ability of the aTBP method in discriminating among animal species is based on the variation of the length of intron III, commonly found in members of the animal β-tubulin gene family that may differ by number. Therefore, the same couple of primers conveniently located at the boundaries of the third intron amplifies, in a typical PCR reaction, a group of fragments that can vary in number and length in each analyzed species. If resolved in a capillary electrophoresis system, they eventually define a species-specific DNA code. The separation resolution is such that peaks/fragments differing from just 1–2 bp can be recognized. Each peak of the electropherogram is defined by sizes, expressed in bp, and by a height, expressed in Relative Fluorescence Units (RFU) values.

A good and paradigmatic example of the level of information that is retrievable by aTBP, when applied to individuals of the same species, is provided in Table 1 with reference to gilthead seabream (*S. aurata*). Sixteen different individuals coming from aquaculture, already characterized for their morphological traits and with a panel of SSRs primers, were analyzed with aTBP together with 6 individuals of the same species purchased in the market and classified by *COI* barcoding. The data reveal the presence of five amplified fragments that are commonly shared among all of the analyzed samples (grey columns in Table 1), In addition, a quite diffuse and interesting intra-species variation, characterized in both subgroups by either missing or supplementary amplicons, likely corresponding to allelic variations, was identified. It is of relevance to note that, with the exception of the 312 bp long fragment amplified from the DNA extracted from three individuals of the Spallanzani group of specimens, the DNA polymorphisms detected at intra-species level are present in both the analyzed groups, likely reflecting ongoing variations in the general gilthead seabream population.

A similar situation was found when analysing 16 samples of the Adriatic sturgeon (*A. naccarrii*). Once more, commonly shared amplicons were found together with intraspecific polymorphisms, as shown in the upper panel of Table 2. Quite remarkably, one of these samples (A10) showed a very different pattern of aTBP amplification, perfectly matching that found in two available samples of white sturgeon (*A. transmontanus*). This reassignment is fully consistent with data previously obtained on the same experimental group with the use of a panel of SSR markers [19].

Data reported in Table 3 more adequately underscore the application of the aTBP method for the discrimination of two different, important and largely commercialized tuna species: red tuna (*Thunnus thynnus*) and yellowfin tuna (*Thunnus albacares*). As can be easily appreciated, the two tuna species show commonly shared amplified aTBP fragments, referable to their genus, and species-specific fragments, two of 255 bp and 778 bp, and one of 282 bp in yellowfin and red tuna, respectively. Once more, both groups are further characterized by the presence of additional intraspecies polymorphic fragments that may be shared or not between the two species.

The ability of aTBP to easily discriminate among different fish species, revealed by the data just presented, motivated us to verify if the method could be used as a simple way to detect

**Table 1. aTBP profile of 22 samples of gilthead seabram (*S. aurata*).**

| Sample | | CE peaks | | | | | | | | | | | | | |
| --- | --- | --- | --- | --- | --- | --- | --- | --- | --- | --- | --- | --- | --- | --- | --- |
| A5 | Size | | 210.4 | | 214.4 | 243.9 | | 284.6 | | | 316.6 | 348.8 | 356.9 | 562.1 | 645.4 |
| | Height | | 31306 | | 22426 | 21955 | | 9556 | | | 15357 | 1519 | 1239 | 4798 | 4774 |
| B5 | Size | 209.0 | 210.0 | 211.1 | 214.6 | 244.0 | | 284.8 | | 312.6 | 315.8 | 349.0 | | 562.4 | 645.9 |
| | Height | 12144 | 13097 | 25174 | 14470 | 15301 | | 4806 | | 8911 | 7919 | 1262 | | 6889 | 2472 |
| C5 | Size | 208.9 | 209.8 | 210.9 | 214.4 | 244.0 | | 284.6 | | | 316.8 | 348.9 | 357.0 | 562.3 | 646.4 |
| | Height | 16768 | 16189 | 30909 | 19393 | 18316 | | 8103 | | | 15216 | 900 | 980 | 5384 | 5089 |
| D5 | Size | | 209.7 | | 214.4 | 244.0 | | 284.6 | 297.3 | | 316.8 | 348.8 | 357.1 | 562.1 | 646.4 |
| | Height | | 30384 | | 30428 | 31122 | | 11928 | 18227 | | 18857 | 1707 | 1579 | 16842 | 10168 |
| E5 | Size | | 209.6 | | 214.2 | 243.8 | | 284.6 | 297.4 | | 316.7 | 348.8 | 357.1 | 562.3 | 646.4 |
| | Height | | 29737 | | 29886 | 32583 | | 13433 | 20264 | | 21847 | 1816 | 1779 | 19978 | 11679 |
| F5 | Size | | | 211.0 | 214.6 | 244.1 | | 284.8 | | | 316.8 | 349.0 | 357.2 | 562.6 | 645.8 |
| | Height | | | 30631 | 12439 | 14485 | | 4736 | | | 15886 | 675 | 556 | 6427 | 2255 |
| G5 | Size | | 210.9 | | 214.5 | 244.0 | | 284.5 | 297.4 | | 317.0 | 348.8 | | 562.0 | 646.3 |
| | Height | | 24018 | | 9643 | 8789 | | 3323 | 5381 | | 5070 | 851 | | 4471 | 1470 |
| H5 | Size | | | 211.0 | 214.6 | 244.0 | | 284.6 | | | 317.0 | 348.9 | 357.2 | 562.5 | 646.8 |
| | Height | | | 15524 | 5955 | 6998 | | 2243 | | | 6551 | 305 | 287 | 2542 | 1971 |
| A6 | Size | | 210.8 | | 214.4 | 244.0 | | 284.6 | 297.3 | | 316.6 | 348.9 | 357.0 | 562.2 | 646.5 |
| | Height | | 28890 | | 11578 | 11432 | | 4860 | 4819 | | 5642 | 710 | 548 | 3836 | 1560 |
| B6 | Size | | 210.8 | | 214.4 | 244.0 | | 284.6 | | 312.5 | | 348.9 | 357.0 | 562.2 | 646.4 |
| | Height | | 31948 | | 7584 | 8826 | | 3609 | | 6286 | | 359 | 372 | 1127 | 1619 |
| C6 | Size | | 210.8 | | 214.4 | 243.9 | | 284.5 | | | 316.7 | 348.8 | 357.1 | 562.2 | 646.3 |
| | Height | | 26365 | | 9916 | 11818 | | 4212 | | | 7978 | 581 | 400 | 1849 | 1049 |
| D6 | Size | | 210.8 | | 214.4 | 243.9 | | 284.6 | | | 316.7 | 348.8 | 356.9 | 562.0 | 645.4 |
| | Height | | 11943 | | 4865 | 4567 | | 2147 | | | 3466 | 289 | 234 | 1212 | 1334 |
| E6 | Size | | 210.8 | | 214.4 | 244.0 | 256.3 | 284.6 | | | 316.6 | 348.7 | 357.1 | 562.5 | 646.3 |
| | Height | | 26742 | | 10695 | 9726 | 6251 | 3716 | | | 5840 | 422 | 567 | 4371 | 1757 |
| F6 | Size | | 210.9 | | 214.5 | 243.9 | | 284.6 | | 312.4 | 315.5 | 348.8 | | 562.1 | 645.5 |
| | Height | | 24948 | | 10458 | 9518 | | 3975 | | 5936 | 4762 | 1018 | | 4322 | 1975 |
| G6 | Size | 208.9 | 210.9 | | 214.4 | 243.9 | | 284.6 | 297.3 | | 316.6 | | 356.9 | 562.3 | 646.4 |
| | Height | 8466 | 16142 | | 5015 | 3607 | | 1466 | 4713 | | 4649 | | 197 | 6864 | 2020 |
| H6 | Size | 209.0 | 210.5 | | | 243.9 | | 284.6 | | | 316.8 | 348.8 | 357.1 | 562.1 | 646.4 |
| | Height | 14288 | 31750 | | 17309 | 15993 | | 7099 | | | 14240 | 995 | 784 | 5823 | 4895 |
| FT49 | Size | 209.0 | 210.1 | 211.1 | 214.6 | 244.1 | | 284.8 | 297.6 | | 317.0 | 349.0 | | 562.4 | 646.8 |
| | Height | 16896 | 19169 | 28429 | 18824 | 17453 | | 6065 | 9577 | | 8942 | 1376 | | 7652 | 3003 |
| FT99 | Size | 208.9 | 209.9 | 211.0 | 214.6 | 244.0 | 256.5 | 284.7 | 297.6 | | | 348.7 | 357.1 | 562.5 | 646.8 |
| | Height | 27505 | 21764 | 30946 | 20052 | 15411 | 12155 | 4287 | 10438 | | | 187 | 925 | 5081 | 2402 |
| FT128 | Size | 208.8 | 209.9 | 210.9 | 214.6 | 244.1 | | 284.8 | 297.6 | | 317.1 | 349.1 | 357.1 | 562.5 | 646.7 |
| | Height | 25141 | 30944 | 7263 | 30616 | 25256 | | 7525 | 16964 | | 15525 | 933 | 675 | 8415 | 3913 |
| FT290 | Size | 209.0 | 209.6 | 210.5 | 214.4 | 244.0 | | 284.6 | 297.3 | | | | 356.9 | 562.2 | 646.6 |
| | Height | 14445 | 28478 | 28478 | 20145 | 17842 | | 4714 | 11055 | | | | 773 | 2247 | 507 |
| FT287 | Size | | 209.8 | 210.4 | 214.4 | 243.9 | | 284.5 | 297.3 | | 316.7 | 348.8 | 357.0 | 562.2 | 646.3 |
| | Height | | 24362 | 31171 | 18013 | 17275 | | 4347 | 9780 | | 7490 | 567 | 589 | 6011 | 1498 |
| FT310 | Size | | 210.0 | 211.1 | 214.6 | 244.1 | 256.6 | 284.7 | 297.5 | | | 348.9 | 357.1 | 562.4 | 646.6 |
| | Height | | 19834 | 29679 | 19140 | 16886 | 11181 | 5112 | 8455 | | | 493 | 447 | 4689 | 2169 |

Size refers to the length, in base pairs, of the amplified TBP fragment; Height refers to the signal intensity, expressed in RFU. A5-H6: samples provided by the Lazzaro Spallanzani Research Institute; FT49-FT310: samples characterized by *COI* sequencing at the University of Siena.

Grey columns show the peaks shared among all the analyzed samples.

**Table 2. aTBP profile of sturgeon (*Acipenser* spp.).**

| | Sample | | CE peaks | | | | | | | | | | | | |
|---|---|---|---|---|---|---|---|---|---|---|---|---|---|---|---|
| Adriatic sturgeon | A9 | Size | 245.0 | 250.4 | 251.2 | 254.3 | 289.7 | 630.9 | 775.6 | | 860.2 | 863.2 | 865.0 | 1013.9 |
| | | Height | 12175 | 9723 | 12043 | 5772 | 7682 | 249 | 2272 | | 391 | 364 | 412 | 1549 |
| | B9 | Size | 245.1 | 250.5 | 251.4 | 254.5 | 290.0 | 631.1 | 775.8 | 853.6 | 860.2 | 863.7 | 865.4 | 1014.1 |
| | | Height | 21322 | 13189 | 21903 | 17113 | 18187 | 870 | 4125 | 851 | 1425 | 784 | 454 | 2563 |
| | C9 | Size | 245.1 | 250.4 | 251.3 | 254.3 | 289.8 | 630.9 | 775.6 | 853.8 | 860.1 | | 865.1 | 1014.0 |
| | | Height | 11809 | 4626 | 7084 | 5467 | 7036 | 598 | 925 | 618 | 733 | | 481 | 1135 |
| | D9 | Size | 245.0 | 250.3 | 251.2 | 254.3 | 289.7 | 630.8 | 775.5 | 852.6 | 860.2 | | 865.2 | 1013.9 |
| | | Height | 9122 | 14802 | 5291 | 4354 | 4024 | 261 | 557 | 538 | 482 | | 128 | 748 |
| | E9 | Size | 245.0 | 250.4 | 251.2 | 254.3 | 289.7 | 630.8 | 775.2 | | 860.2 | 863.1 | 864.9 | 1013.9 |
| | | Height | 11901 | 9286 | 11547 | 5505 | 7441 | 212 | 2284 | | 263 | 350 | 600 | 1479 |
| | F9 | Size | 244.6 | 250.0 | 251.2 | 254.3 | 289.6 | | 775.4 | | 860.2 | 863.0 | 865.0 | 1013.7 |
| | | Height | 29308 | 29415 | 11547 | 29563 | 32188 | | 13497 | | 2866 | 3175 | 2505 | 8511 |
| | G9 | Size | 245.1 | 250.5 | 251.4 | 254.4 | 290.0 | 631.1 | 775.9 | 853.2 | 860.5 | 863.8 | 865.6 | 1014.2 |
| | | Height | 28093 | 10014 | 17103 | 12533 | 18004 | 792 | 3808 | 472 | 2209 | 649 | 535 | 2412 |
| | H9 | Size | 245.1 | 250.5 | 251.4 | 254.5 | 290.0 | 631.2 | 775.9 | | | 863.7 | 865.5 | 1014.5 |
| | | Height | 21460 | 13604 | 16855 | 7033 | 14811 | 413 | 2481 | | | 966 | 773 | 1547 |
| | B10 | Size | 245.1 | 250.4 | 251.3 | 254.4 | 289.9 | 631.2 | 775.8 | | 860.8 | 863.8 | | 1014.7 |
| | | Height | 26052 | 13531 | 24414 | 15436 | 22283 | 1547 | 3387 | | 1281 | 1231 | | 2272 |
| | C10 | Size | 245.0 | 250.4 | 251.2 | 254.3 | 289.8 | 630.8 | 775.5 | | 860.2 | 863.1 | | 1013.8 |
| | | Height | 6318 | 5740 | 9844 | 6722 | 7374 | 173 | 1195 | | 637 | 343 | | 625 |
| | D10 | Size | 245.0 | 250.4 | 251.2 | 254.2 | 289.7 | 630.9 | 775.4 | 853.0 | 860.2 | | 865.2 | 1014.0 |
| | | Height | 7802 | 2364 | 3730 | 2900 | 2078 | 92 | 248 | 404 | 281 | | 367 | 211 |
| | E10 | Size | 245.1 | 250.5 | 251.3 | 254.4 | 290.0 | 631.1 | 776.0 | | 860.6 | | 864.8 | 1014.6 |
| | | Height | 7883 | 5876 | 8189 | 3158 | 6151 | 386 | 723 | | 254 | | 191 | 504 |
| | F10 | Size | 245.0 | 250.4 | 251.2 | 254.4 | 289.8 | 630.9 | 775.5 | | 860.3 | | 864.8 | 1014.1 |
| | | Height | 6602 | 6250 | 6427 | 2409 | 5185 | 388 | 1024 | | 235 | | 502 | 643 |
| | G10 | Size | 245.1 | | 251.3 | 254.4 | 290.0 | 631.1 | 775.8 | 852.0 | 860.4 | | 865.3 | 1014.2 |
| | | Height | 12875 | | 11766 | 8299 | 9505 | 953 | 1264 | 790 | 1114 | | 741 | 574 |
| | H10 | Size | 245.0 | | 251.2 | 254.2 | 289.8 | 631.0 | 775.6 | 853.2 | 859.9 | | | 1014.4 |
| | | Height | 7699 | | 5837 | 4089 | 4713 | 312 | 461 | 309 | 395 | | | 311 |

(*Continued*)

**Table 2.** (Continued)

| Sample | | | 251.3 | 252.3 | | 255.4 | 289.5 | 297.7 | 760.3 | 795.8 | 816.3 | 902.9 |
|---|---|---|---|---|---|---|---|---|---|---|---|---|
| White sturgeon | A10 | Size | 251.3 | 252.3 | | 255.4 | 289.5 | 297.7 | 760.3 | 795.8 | 816.3 | 902.9 |
| | | **Height** | **7998** | **9431** | | **2598** | **6235** | **1270** | **1356** | **1531** | **691** | **486** |
| | FT274 | Size | 251.3 | 252.3 | 254.4 | 255.3 | 289.4 | 297.7 | 760.3 | 795.8 | 816.1 | 902.2 |
| | | Height | 5841 | 2923 | 673 | 1440 | 4419 | 1454 | 1636 | 1026 | 1024 | 658 |
| | FT284 | Size | 251.3 | 252.3 | | 255.4 | 289.4 | 297.7 | 760.2 | 795.7 | 816.0 | 902.7 |
| | | Height | 6761 | 10520 | | 2375 | 6673 | 1156 | 1676 | 1141 | 390 | 493 |

Analysis of 16 samples of Adriatic sturgeon provided by the Lazzaro Spallanzani Research Institute and 2 samples of white sturgeon purchased in the market. For each analysed sample, numerical values refer to the sizes of the amplified TBP fragment, in bp, and to the signal height in RFU, respectively.

Light grey columns: peaks commonly shared between the two species.

Dark grey columns with white numbers: species-specific diagnostic fragments.

Bold fonts: sample of the first set erroneously classified as *A. naccarii*.

**Table 3. aTBP profiles of tunafish (*Thunnus* spp.).**

| | Sample | Size values | | | | | | | | | | | | |
|---|---|---|---|---|---|---|---|---|---|---|---|---|---|---|
| Red tuna | A11 | 219.3 | 220.7 | | 230.5 | 253.2 | | 282.9 | | 387.0 | 519.5 | | 796.3 | |
| | B11 | 219.7 | 221.1 | | 230.6 | | | 283.1 | | 387.2 | 519.8 | | 796.6 | |
| | C11 | 219.1 | 220.5 | | 230.3 | | | 282.8 | | 387.0 | 519.5 | | 796.3 | |
| | D11 | 219.1 | 220.5 | | 230.3 | | | 282.8 | 373.2 | 387.0 | 519.5 | | 797.2 | |
| | E11 | 219.7 | 221.2 | | 230.7 | | | 283.2 | | 387.2 | 520.0 | | 796.7 | |
| | F11 | 219.1 | 220.5 | | 230.2 | | | 282.8 | | 387.1 | 519.7 | | 796.1 | |
| | G11 | 219.7 | 221.0 | | 230.6 | | | 283.2 | | | 519.8 | | 797.4 | |
| | H11 | 219.7 | 221.0 | | 230.6 | | | 283.1 | | 387.3 | 519.9 | | 797.0 | |
| | A12 | 219.7 | 221.1 | | 230.7 | | | 283.2 | 373.5 | 387.2 | 519.9 | | 796.9 | |
| | B12 | 219.7 | 221.1 | | 230.6 | | | 283.2 | 373.0 | 387.4 | 519.8 | | 797.8 | 804.7 |
| | C12 | 218.9 | 220.4 | | 230.2 | | | 282.7 | | 386.9 | 519.5 | | 796.3 | |
| | D12 | 219.9 | 221.3 | | 230.7 | | | 283.4 | | 387.5 | 520.2 | | 797.4 | |
| | E12 | 219.9 | 221.3 | | 230.8 | | | 283.3 | | 387.5 | 520.1 | | 797.4 | |
| | F12 | 219.7 | 221.1 | | 230.6 | | | 283.0 | | 387.3 | 519.9 | | | 805.3 |
| | G12 | 219.7 | 221.1 | | 230.6 | | | 283.1 | 373.5 | | 519.9 | | | 804.6 |
| | H12 | 219.7 | 221.1 | | 230.6 | | | 283.1 | | 387.1 | 520.0 | | 797.0 | 805.7 |
| Yellowfin tuna | FT95 | | 221.1 | | 230.7 | | 255.4 | | | 387.5 | 519.9 | 778.3 | | |
| | FT201 | | 220.6 | | 230.7 | | 255.4 | | 373.7 | | 519.9 | 777.9 | 796.7 | |
| | FT189 | 219.5 | 221.1 | | 230.4 | 253.4 | 255.4 | | 373.6 | 387.4 | 519.8 | 778.1 | | |
| | FT267 | 219.9 | 221.1 | | 230.6 | | 255.4 | | 373.6 | 387.4 | 519.8 | 778.4 | | |
| | FT308 | | 220.6 | | 230.6 | | 255.4 | | | 387.5 | 519.8 | 778.4 | | |
| | FT357 | 220.2 | 221.4 | 224.8 | 230.8 | | 255.6 | | | 387.6 | 520.0 | 778.5 | 797.2 | |

Analysis of 16 samples of red tuna provided by the Lazzaro Spallanzani Research Institute and 6 samples of yellowfin tuna previously characterized at the University of Siena. Only numerical values referring to the sizes, in bp, of the ampiflied TBP fragments are shown.

Light grey columns: commonly shared peaks.

Dark grey columns with white numbers: species-specific diagnostic fragments.

fraud and substitutions, frequently reported, and to a vast scale, in the fisheries market [25]. To this purpose, we analyzed and compared the aTBP profile of two fish species, pangasius (*P. hypophthalmus*) and European seabass (*D. labrax*), because the latter is often replaced by the former when commercialized as fillets or canned food.

Fig 2 readily shows how the two species look completely different from each other when their corresponding aTBP profiles are compared. Not a single amplified fragment is shared among those that are species- specific. As shown, in case of a suspected substitution, this difference can be conveniently revealed by a simple electrophoresis run of the amplified fragments in an agarose gel.

The Salmonidae is a particularly relevant fish family often studied with reference to multiple important issues such as variations in response to climate changes, reproductive habits and parentage recognition, metabolic species-specific features, and, of course, market traceability. Table 4 shows the data obtained applying the aTBP method to individuals of four different salmonid species: carpione trout (*Salmo carpio*), an endemic species of the Garda lake in Italy, brown trout fario (*Salmo trutta* f. *fario*), Atlantic salmon (*Salmo salar)* and rainbow trout (*Oncorhynchus mykiss)*, belonging to two genera of the family and present in natural environments as different as ocean or fresh water.

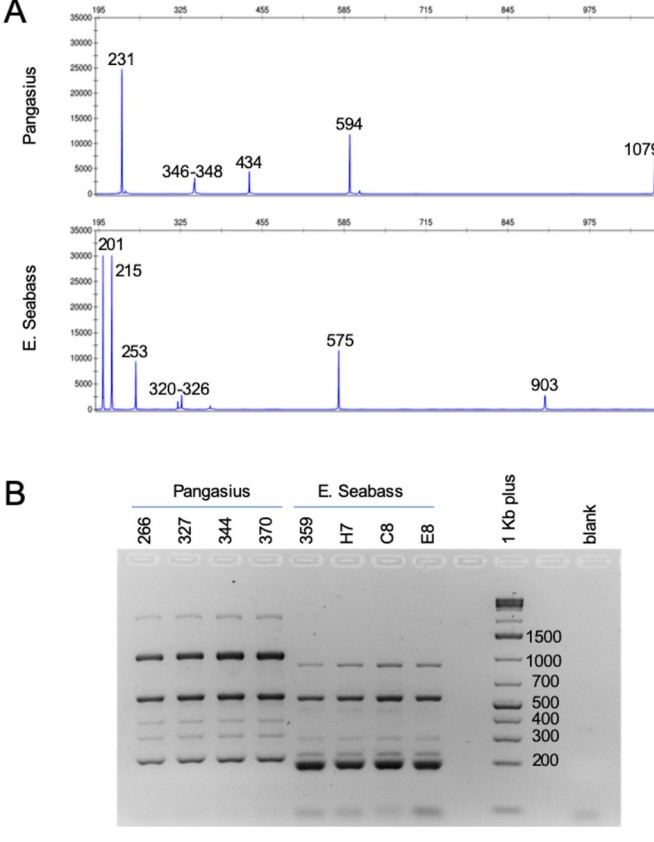

**Fig 2. Comparison between seabass and pangasius aTBP.** A) electropherograms obtained from pangasius (top panel) and European seabass (bottom panel) respectively. Reported numbers on top of the peaks refer to the sizes of the amplified fragments. A typical example is shown. Complete data are reported in S1 Table. B) Separation in agarose gel of the aTBP fragments amplified from pangasius or seabass samples. The numbers and letters above the agarose gel indicate the samples analyzed. The numbers next to the "1kb plus" marker indicate the molecular weights of each fragment.

With the premise that intraspecific polymorphisms, present also in these groups, have been reduced to a minimum to set up a consultable table of immediately appreciable results, Table 4 delivers several useful information. First, the amplified fragments can be individually assigned to different taxonomic ranks, starting with the 220 bp long amplicon that is attributable to the Salmonidae family since it is present in all the samples we have analyzed. The three species belonging to the *Salmo* genus also share five common aTBP amplified fragments (219, 228, 259, 289 and 330 bp) while each single species is characterized by the presence of a small yet variable number of clearly specific amplification products, shown in the dark grey columns of Table 4. Additional similarities, such as those between carpione trout and brown trout fario, are notable (boxed columns).

Similarity between these two species, indicating their more recent separation, was further confirmed by the Principal Component Analysis (PCA) of Fig 3, where the four salmonid species are distributed in three major directions for a cumulative contribution of the first two principal components that explains 76% of the total variance. The complete data set used for PCA is provided in S2 Table.

**Table 4. aTBP analysis of the Salmonidae family.**

| Species | Sample | | | | | | | | | | Size values | | | | | | | | | |
|---|---|---|---|---|---|---|---|---|---|---|---|---|---|---|---|---|---|---|---|---|
| Carpione | C3 | | 214.7 | 219.9 | 220.8 | 223.8 | | | 228.9 | 230.9 | 234.0 | 236.3 | 259.3 | 289.1 | | 303.8 | | 330.6 | | 338.4 |
| | D3 | | 214.7 | 219.9 | 220.8 | 223.7 | | | 228.8 | 230.9 | 233.9 | 236.3 | 259.3 | 289.0 | | 303.8 | | 330.6 | | 338.4 |
| | E3 | 213.7 | 214.7 | 219.9 | 220.8 | 223.8 | | | 228.9 | 230.8 | 234.0 | 236.3 | 259.4 | 289.0 | | 303.8 | | 330.6 | | 338.4 |
| | F3 | | 214.7 | 219.9 | 220.7 | 223.7 | | | 228.8 | | 233.9 | 236.3 | 259.4 | 289.0 | | 303.8 | | 330.6 | | 338.4 |
| | G3 | | 214.7 | 219.9 | 220.7 | 223.7 | | | 228.8 | | 233.9 | 236.2 | 259.4 | 289.1 | | 303.8 | | 330.6 | | 338.4 |
| | A4 | | 214.6 | 219.7 | 220.5 | 223.5 | | | 228.7 | 230.9 | 233.8 | 236.1 | 259.2 | 288.8 | | 303.7 | | 330.3 | | 338.1 |
| | B4 | | 214.7 | 219.9 | 220.7 | 223.6 | | | 228.8 | 230.9 | | 236.2 | 259.3 | 289.0 | | 303.8 | | 330.5 | | 338.3 |
| | C4 | 213.7 | 214.7 | 219.9 | 220.8 | 223.7 | | | 228.9 | 230.8 | 234.0 | 236.2 | 259.3 | 289.0 | | 303.9 | | 330.5 | | 338.3 |
| | D4 | | 214.8 | 219.9 | 220.8 | 223.7 | | | 228.9 | 230.9 | | 236.2 | 259.3 | 289.0 | | 303.8 | | 330.5 | | 338.3 |
| | E4 | 213.5 | 214.5 | 219.7 | 220.5 | 223.5 | | | 228.6 | 230.6 | | 236.0 | 259.2 | 288.9 | | 303.6 | | 330.3 | | 338.1 |
| | F4 | 213.5 | 214.6 | 219.7 | 220.6 | 223.6 | | | 228.6 | 230.7 | | 236.1 | 259.1 | 288.8 | | 303.6 | | 330.4 | | 338.2 |
| | G4 | | 214.6 | 219.7 | 220.5 | 223.5 | | | 228.7 | 230.6 | | 236.1 | 259.2 | 288.8 | | 303.6 | | 330.3 | | 338.1 |
| Brown trout fario | 5 | 213.3 | | 219.6 | 220.4 | 223.5 | | | 228.5 | | 233.6 | 236.0 | 259.1 | 288.7 | | | 305.4 | 330.2 | | 338.1 |
| | 6 | 213.5 | | 219.7 | 220.6 | 223.6 | | | 228.7 | | 233.7 | 236.1 | 259.2 | 288.8 | | | 305.5 | 330.4 | | 338.3 |
| | 8 | 213.3 | | 219.6 | 220.4 | 223.4 | | | 228.5 | | 233.5 | 235.9 | 259.1 | 288.6 | | | 305.4 | 330.3 | | 338.1 |
| | 10 | 213.5 | | 219.7 | 220.6 | 223.6 | | | 228.7 | | 233.8 | 236.1 | 259.2 | 288.9 | | | 305.6 | 330.5 | | 338.2 |
| | 11 | 213.5 | | 219.7 | 220.5 | 223.6 | | | 228.7 | | 233.8 | 236.2 | 259.2 | 288.8 | | | 305.6 | 330.4 | | 338.3 |
| | 12 | 213.5 | | 219.7 | 220.5 | 223.5 | | | 228.6 | | 233.8 | 236.1 | 259.2 | 288.8 | | | 305.5 | 330.4 | | 338.2 |
| | 13 | 213.4 | | 219.6 | 220.4 | 223.4 | | | 228.5 | | 233.6 | 235.9 | 259.1 | 288.7 | | | 305.3 | 330.3 | | 338.1 |
| | 14 | 213.3 | | 219.6 | 220.4 | 223.5 | | | 228.7 | | 233.6 | 235.9 | 259.1 | 288.7 | | | 305.5 | 330.3 | | 338.0 |
| | 15 | 213.3 | | 219.5 | 220.4 | 223.4 | | | 228.5 | | 233.6 | 236.0 | 259.1 | 288.7 | | | 305.4 | 330.3 | | 338.0 |
| | 17 | 213.3 | | 219.6 | 220.4 | 223.5 | | | 228.6 | | 233.6 | 236.0 | 259.1 | 288.7 | | | 305.3 | 330.2 | | 338.1 |
| | 19 | 213.3 | | 219.6 | 220.4 | 223.5 | | | 228.6 | | 233.6 | 236.0 | 259.1 | 288.7 | | | 305.4 | 330.2 | | 338.1 |
| | 22 | 213.6 | | 219.8 | 220.6 | 223.6 | | | 228.8 | | 233.8 | 236.1 | 259.3 | 288.8 | | | 305.5 | 330.4 | | 338.2 |
| Salmon | FT1 | 213.3 | | 219.3 | 220.3 | | 224.5 | 225.8 | 228.4 | 230.5 | | | 259.1 | 288.4 | 290.5 | | | 330.0 | 331.4 | |
| | FT113 | 213.3 | | 219.3 | 220.3 | | 224.5 | 225.8 | 228.5 | 230.6 | | | 259.1 | 288.5 | 290.6 | | | 330.0 | 331.4 | |
| | FT198 | 213.3 | | 219.4 | 220.4 | | 224.5 | 225.9 | 228.5 | 230.6 | | | 259.1 | 288.5 | 290.6 | | | 330.0 | 331.4 | |
| | FT203 | 213.3 | | 219.3 | 220.4 | | 224.5 | 225.8 | 228.5 | 230.6 | | | 259.1 | 288.5 | 290.6 | | | 330.0 | 331.5 | |
| | FT252 | 213.3 | | 219.3 | 220.4 | | 224.6 | 225.9 | 228.5 | 230.6 | | | 259.2 | 288.5 | 290.6 | | | 330.1 | 331.4 | |
| | FT273 | 213.3 | | 219.3 | 220.4 | | 224.4 | 225.8 | 228.5 | 230.6 | | | 259.1 | 288.6 | 290.7 | | | 330.0 | 331.4 | |
| | FT296 | 213.3 | | 219.4 | 220.4 | | 224.4 | 225.8 | 228.5 | 230.6 | | | 259.1 | 288.5 | 290.6 | | | 329.9 | 331.4 | |
| | FT368 | 213.3 | | 219.3 | 220.4 | | 224.5 | 225.8 | 228.8 | 230.6 | | | 259.1 | 288.5 | 290.6 | | | 330.0 | 331.4 | |

*(Continued)*

**Table 4.** (Continued)

| Rainbow trout | | | | | | | | | | | | |
|---|---|---|---|---|---|---|---|---|---|---|---|---|
| A1 | 213.2 | 220.5 | 223.6 | 224.8 | 225.9 | 232.8 | 258.3 | 290.5 | 309.4 | 324.5 | 334.0 |
| D1 | 212.9 | 220.5 | 223.5 | 224.6 | 226.0 | | 258.1 | 290.7 | 309.1 | 324.2 | 333.9 |
| E1 | 213.0 | 220.4 | 223.5 | 224.6 | 225.7 | 232.6 | 258.1 | 290.4 | 309.1 | 324.3 | 333.8 |
| G1 | 212.4 | 220.3 | 223.4 | 224.5 | 225.7 | | 258.2 | 290.4 | 309.0 | 324.3 | 333.7 |
| E2 | 213.1 | 220.5 | 223.6 | 224.8 | 225.8 | 232.8 | 258.4 | 290.6 | 309.4 | 324.5 | 334.0 |
| F2 | 213.3 | 220.5 | 223.6 | 224.9 | 225.9 | 232.8 | 258.3 | 290.7 | 309.3 | 324.5 | 333.9 |
| G2 | 212.9 | 220.3 | 223.4 | 224.6 | 225.9 | 232.6 | 258.2 | 290.5 | 309.2 | 324.2 | 333.8 |
| H2 | 213.0 | 220.4 | 223.4 | 224.6 | 225.7 | 232.7 | 258.2 | 290.4 | 309.1 | 324.3 | 333.9 |
| FT119 | 212.6 | 220.1 | 223.1 | 224.6 | 225.9 | 232.7 | 257.8 | 290.3 | 309.2 | 324.4 | 333.7 |
| FT22 | 213.0 | 220.4 | 223.5 | 224.7 | 225.7 | 232.7 | 258.2 | 290.6 | 309.1 | 324.2 | 333.7 |
| FT262 | 213.1 | 220.4 | 223.5 | 224.6 | 226.0 | 232.6 | 258.2 | 290.5 | 309.1 | 324.2 | 333.8 |
| FT334 | 212.5 | 220.4 | 223.5 | 224.7 | 225.7 | | 258.2 | 290.6 | 309.2 | 324.2 | 333.9 |

Size values

| Species | Sample | | | | | | | | | | |
|---|---|---|---|---|---|---|---|---|---|---|---|
| Carpione | C3 | 358.6 | | 374.5 | 388.9 | 412.5 | 414.3 | 747.0 | 790.1 |
| | D3 | 358.6 | | 374.4 | 388.9 | 412.5 | 414.3 | 747.0 | 790.1 |
| | E3 | 358.7 | | 374.5 | 388.9 | 412.6 | 414.3 | 746.9 | 790.2 |
| | F3 | 358.6 | | 374.5 | 388.9 | 412.4 | 414.3 | 747.0 | 790.1 |
| | G3 | 358.6 | | 374.5 | 388.9 | 412.5 | 414.3 | 747.0 | 790.1 |
| | A4 | 358.4 | | 374.3 | 388.6 | 412.2 | 414.0 | 746.3 | 789.7 |
| | B4 | 358.6 | 363.4 | 374.5 | 388.8 | 412.5 | 414.3 | 746.9 | 789.9 |
| | C4 | 358.6 | | 374.5 | 388.9 | 412.4 | 414.3 | 746.8 | 790.3 |
| | D4 | 358.5 | | 374.5 | 388.8 | 412.5 | 414.3 | 746.8 | 790.1 |
| | E4 | 358.4 | 363.3 | 374.3 | 388.6 | 412.3 | 414.0 | 746.4 | 789.7 |
| | F4 | 358.4 | 363.3 | 374.2 | 388.7 | 412.3 | 414.1 | 746.3 | 789.5 |
| | G4 | 358.4 | 363.2 | 374.2 | 388.6 | 412.3 | 414.1 | 746.4 | 789.7 |
| Brown trout fario | 5 | 357.2 | 363.1 | | 388.5 | 412.2 | 414.0 | 746.1 | 789.5 |
| | 6 | 357.5 | 363.3 | | 388.7 | 412.3 | 414.1 | 746.3 | 789.8 |
| | 8 | 357.2 | 363.1 | | 388.5 | 412.2 | 413.8 | 746.1 | 789.6 |
| | 10 | 357.4 | 363.2 | | 388.7 | 412.4 | 414.1 | 746.2 | 789.7 |
| | 11 | 357.4 | 363.3 | | 388.6 | 412.4 | 414.1 | 746.6 | 789.9 |
| | 12 | 357.4 | 363.2 | | 388.7 | 412.4 | 414.0 | 746.7 | 789.9 |
| | 13 | 357.2 | 363.1 | | 388.6 | 412.1 | 414.0 | 746.2 | 789.6 |
| | 14 | 357.3 | 363.2 | | 388.6 | 412.2 | 413.9 | 746.1 | 789.7 |
| | 15 | 357.3 | 363.2 | | 388.5 | 412.2 | 413.9 | 746.1 | 789.4 |
| | 17 | 357.3 | 363.1 | | 388.5 | 412.2 | 414.0 | 746.1 | 789.5 |
| | 19 | 357.2 | 363.1 | | 388.5 | 412.2 | 414.2 | 746.1 | 789.5 |
| | 22 | 357.4 | 363.2 | | 388.8 | 412.3 | 414.0 | 746.4 | 789.8 |

(*Continued*)

**Table 4.** (Continued)

| | | ~340 | ~347 | ~356 | ~358 | ~361 | ~388 | ~392 | ~393 | ~398 | ~414 | ~476 | ~742 | ~745 | ~858 | ~860 | ~1013 |
|---|---|---|---|---|---|---|---|---|---|---|---|---|---|---|---|---|---|
| Salmon | FT1 | 340.7 | | 356.2 | | | 388.5 | 391.9 | | | 414.0 | 476.3 | | 745.5 | 858.2 | | |
| | FT113 | 340.7 | 347.5 | | 358.1 | | 388.5 | 392.0 | | | 414.0 | 476.3 | | 745.4 | | | |
| | FT198 | 340.8 | | 356.2 | | | 388.5 | 392.1 | | | 414.0 | 476.3 | | 745.4 | 858.1 | | |
| | FT203 | 340.7 | 347.5 | 356.1 | | | 388.5 | 392.0 | | | 414.1 | 476.3 | | 745.6 | 858.2 | | |
| | FT252 | 340.8 | | 356.2 | 358.2 | | 388.5 | 392.1 | | | 414.0 | 476.4 | | 745.4 | 858.2 | | |
| | FT273 | 340.8 | 347.5 | 356.3 | | | 388.5 | 392.0 | | | 414.1 | 476.4 | | 745.4 | 858.1 | | |
| | FT296 | 340.8 | 347.6 | 356.3 | | | 388.4 | 392.0 | | | 414.1 | 476.4 | | 745.7 | 858.0 | | |
| | FT368 | 340.7 | | 356.2 | 358.2 | | 388.5 | 392.1 | | | 414.0 | 476.4 | | 745.5 | 858.2 | | |
| Rainbow trout | A1 | | | | | 361.5 | 388.2 | | 393.5 | 398.5 | 414.0 | | 742.4 | | | 860.3 | 1013.6 |
| | D1 | | | | | 361.5 | | | 393.2 | 398.4 | 414.0 | | 742.3 | | | 860.0 | 1013.4 |
| | E1 | | | | | 361.3 | 388.0 | | 393.1 | 398.2 | 413.8 | | 742.0 | | | 859.7 | 1013.2 |
| | G1 | | | | | 361.3 | 387.9 | | 393.1 | 398.3 | 413.8 | | 741.9 | | | 859.6 | 1013.1 |
| | E2 | | | | | 361.5 | 388.2 | | 393.2 | 398.5 | 414.1 | | 742.6 | | | 860.2 | 1013.9 |
| | F2 | | | | | 361.5 | 388.2 | 391.3 | 393.5 | 398.5 | 414.0 | | 742.6 | | | 860.3 | 1013.7 |
| | G2 | | | | | 361.4 | 388.0 | | 393.2 | 398.2 | 413.8 | | 742.1 | | | 859.9 | 1013.2 |
| | H2 | | | | | 361.3 | | | 393.1 | 398.2 | 413.8 | | 742.0 | | | 859.9 | 1013.5 |
| | FT119 | | | | | 361.4 | | | 393.3 | | 413.9 | | 742.1 | | | 860.0 | 1013.2 |
| | FT262 | | | | | 361.3 | 388.0 | | 393.2 | 398.3 | 413.9 | | 742.1 | | | 859.8 | 1013.4 |
| | FT334 | | | | | 361.4 | 388.0 | | 393.2 | | 414.0 | | 742.1 | | | 859.9 | 1013.4 |

With the exception of *salmon*, for which just 8 samples were available, a selection of 12 individuals out of the total for each of the other three salmonid species, is shown. The complete dataset is provided in S2 Table.

Intermediate grey columns: Salmonidae family; Light grey columns: Salmonidae family; Light grey columns: *Salmo* genus; Dark grey columns with white numbers: species; boxed numbers: peaks/amplicons common to the *Carpione trout* and the *brown trout fario*.

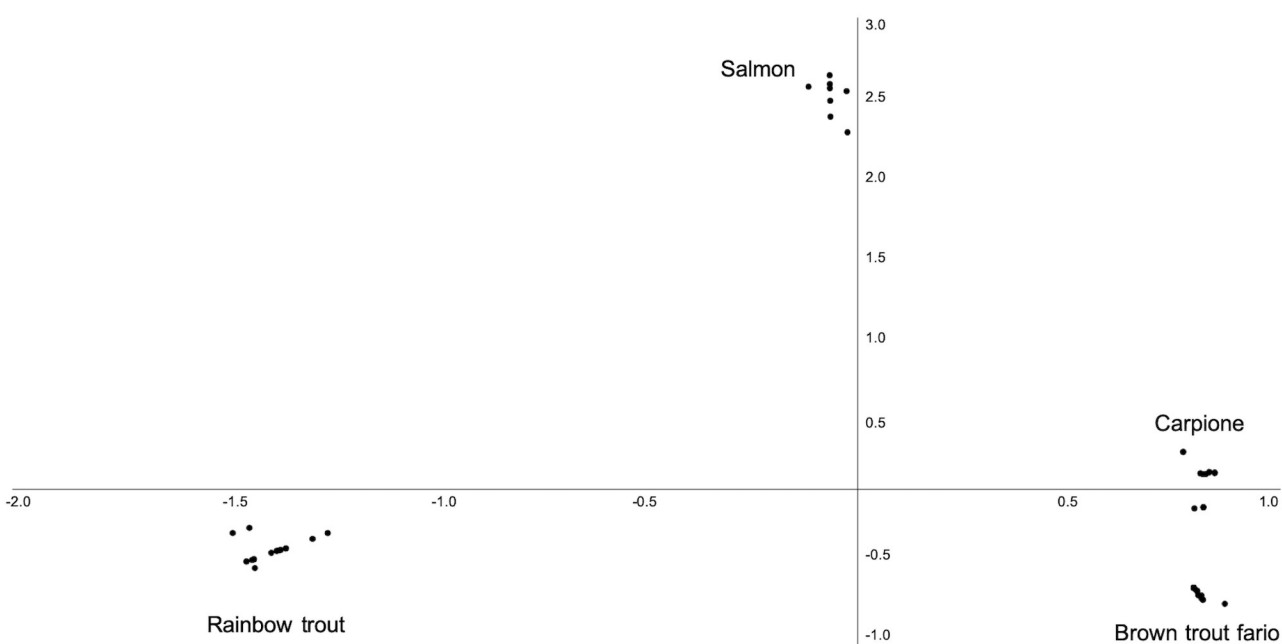

**Fig 3. PCA of the Salmonidae diversity based on aTBP.** The presence-absence matrix obtained by scoring the TBP markers was used to explore the distribution of four fish species belonging the Salmonidae family. The first two principal components explain the 58 and 18% of the variance, respectively.

Overall, the data shown indicate that the aTBP method can be easily and conveniently used to monitor variations occurring at different taxonomic ranks, providing a useful and very versatile tool for different kind of investigations.

## Discussion

This paper presents evidence in favour of the use of the aTBP method for the genetic characterization of fish at different taxonomic levels and for different purposes. We have demonstrated that using a single PCR-based reaction with the same pair of primers, the TBP method can amplify from the genome of any fish sample a number of fragments that delineate a specific DNA profile, or barcode. The aTBP amplification products of a single barcode can then be sequentially attributed to the family, genus, species and subspecies categories. In its essence, aTBP adds to the two fundamental features of an ideal DNA barcode: high taxonomic coverage and high interspecific resolution. Thus, with aTBP recognition of subspecies polymorphisms become simpler and more efficient providing immediate data, with no need for sequencing or necessary prior knowledge of the target sequences. The power of the discrimination of the aTBP genomic profiling method is also shown to be unaffected by ploidy since sturgeon and salmonid species, known polyploidys [26, 27], can be easily distinguished. In fact, the two sturgeon species we have analysed, *A. naccarii and A. transmontanus*, are natural octaploid with 240–264 chromosomes. Due to the high level of fragments resolution granted by CE (1–2 bp), aTBP is expected to perform well also in presence of higher ploidy and chromosome numbers. Problems may arise in the reading of the electropherogram output that can become complex for the presence of numerous peaks. A software that can help in the fast recognition of the output is presently under development. Finally, aTBP is a functional and nuclear-based molecular marker. All these features may offer new opportunities to studies that are performed in diverse

fields of investigation. The exception is molecular taxonomy where a long term, well established, rapidly diffused and internationally supported method based on the sequencing of the mitochondrial *COI* gene has provided the deposition of more than 80.000 barcoding sequences corresponding to approximately 8.000 different fish species. Nevertheless, as also shown in this paper, since aTBP substantially confirm *COI* data, it may be useful when species assignment, based on *COI*, is uncertainly relying on minimal SNPs differences.

This stated, the use of aTBP for identification, authentication and detection of fish species in food samples is quite appropriate and particularly suitable for all those laboratories that are not equipped with demanding sequencing facilities. As a classical DNA barcoding, aTBP can be applied to a high number of species, characterized by a large spectrum of variation. Differently from a classical DNA barcoding, the aTBP primers are effective independently from the taxonomic rank while *COI* primers must be often optimized for the successful use at ranks higher than species. In addition, aTBP can be used for detecting subspecies populations and local varieties. Anyhow, both applications, aTBP and classical DNA barcoding, are particularly suitable for seafood traceability, especially when transformation processes make morphological inspection impossible for fillets, frozen and canned foods, fostering frauds and substitutions. These irregularities could be easily uncovered by the detection of the aTBP species-specific diagnostic peaks as well as the visualization, even in a very simple agarose gel, of very diverse patterns of amplification as here shown for pangasius versus seabass (Fig 2). aTBP can also be of help for assessing variation in a natural population, a major goal in the field of evolutionary biology. To this regard, it is of interest to highlight the finding of a hierarchical distribution that assigns specific aTBP amplification fragments to different taxonomic ranks, as observed in *Thunnus*, *Acipenser* and Salmonidae. It looks like evolution has left molecular traces of its action in the introns of tubulin, from family down to species, and the presence of intra specific subpopulations, characterized by the sharing of few polymorphisms, promise to be a renovated handle for monitoring future evolutions. Since these intraspecific changes in allele frequency can be easily scored, they provide useful information on the overall structure of populations with respect to vulnerability, or resilience, in response to environmental changes and in natural selection constraints. Unique responses often are associated with mutations in genomic regions related to metabolic, developmental, immunogenic and physiological processes. aTBP genomic profiling is based on a functional marker, that is tubulin, since long related to cold response because of the identification of cold-inducible promoters and aminoacid changes exclusively present in the α- and β-tubulin moieties of the Antarctic fishes. Thus it is reasonable to consider the aTBP genomic profiling as a useful tool that can further our understanding of changes in fish genotypes and variations in population fitness.

Another field of possible and useful application of the aTBP method is the potential contribution to our understanding of the role that natural or anthropogenic hybridization and sexual competition play in genetic diversity including breeding among native and introduced species. For example, aTBP could be used for identifying preferential occupation of spawning grounds by a given species as well as recognition of the breeding system and parental assignment. Since the aTBP is a nuclear-based codominant marker, its usage may favor the recognition of hybrids already present in the F1 generation, rather than the F2 populations as is commonly practiced by the use of the mitochondrial, maternally inherited *COI* gene. In summary, understanding the processes underlying diversification can aid in formulating appropriate conservation management plans that will help to maintain the evolutionary potential of taxa, particularly under human-induced activities and climate changes.

Under most practical terms, aTBP is a simple and quick technique, based on a single PCR reaction and the resolution of the amplified fragments by electrophoresis, that may take few hours for an easy recognition on an agarose gel. Several samples can be concomitantly

analyzed, 24 a day in our experience, providing consistent and reproducible genomic profiles that assist in the characterization of the genetic variation of the investigated species. A possible further improvement could be obtained by combining aTBP amplification to High Resolution Melting, as recently done for a combination of different plant DNA barcodes [28]. Also, efforts are in place to establish a practical aTBP data base with the help of Institutions and fishery companies. In conclusion, aTBP should be considered as valuable new tool of genetic investigation in fish for its simplicity of use, good costs/effectiveness ratio, usefulness in different fields of application and wide taxonomic coverage.

## Supporting information

**S1 Table. Seabass and pangasius aTBP analysis, complete dataset.**
(XLSX)

**S2 Table. Salmonidae aTBP analysis, complete dataset used for PCA.**
(XLSX)

**S1 Data. COI sequences, aligned and unaligned, of different fish species.**
(TXT)

**S1 Fig.**
(TIF)

## Acknowledgments

We wish to thank Dr. Luca Braglia for his contribution on PCA. We also want to acknowledge Prof. Khidir Hilu of Virginia Tech, USA and Prof. Sara Patterson, Emeritus at Wisconsin University USA for their critical reading of the manuscript.

This work was partially supported by the Future Home for Future Communities (FHfFC) project funded by Regione Lombardia. There was no additional external funding received for this study and the funders had no role in study design, data collection and analysis, decision to publish, or preparation of the manuscript.

## Author Contributions

**Conceptualization:** Diego Breviario.

**Data curation:** Silvia Gianì, Silvia Silletti, Laura Morello, Giacomo Spinsanti, Katia Parati.

**Formal analysis:** Laura Morello.

**Funding acquisition:** Diego Breviario.

**Investigation:** Silvia Gianì, Silvia Silletti, Giacomo Spinsanti, Katia Parati.

**Methodology:** Silvia Gianì, Silvia Silletti, Floriana Gavazzi.

**Project administration:** Diego Breviario.

**Resources:** Diego Breviario.

**Software:** Floriana Gavazzi.

**Validation:** Laura Morello.

**Writing – original draft:** Diego Breviario.

**Writing – review & editing:** Floriana Gavazzi, Laura Morello.

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
