## [Decision Letter · Decision Letter 0]

22 Apr 2020

PONE-D-20-08308

aTBP: a versatile tool for fish genotyping

PLOS ONE

Dear Dr. Breviario,

Thank you for submitting your manuscript to PLOS ONE. After careful consideration, we feel that it has merit but does not fully meet PLOS ONE’s publication criteria as it currently stands. Therefore, we invite you to submit a revised version of the manuscript that addresses the points raised during the review process.

We would appreciate receiving your revised manuscript by Jun 06 2020 11:59PM. To enhance the reproducibility of your results, we recommend that if applicable you deposit your laboratory protocols in protocols.io, where a protocol can be assigned its own identifier (DOI) such that it can be cited independently in the future. For instructions see: http://journals.plos.org/plosone/s/submission-guidelines#loc-laboratory-protocols

We look forward to receiving your revised manuscript.

Kind regards,

Tzen-Yuh Chiang

Academic Editor

PLOS ONE

Journal Requirements:

3. Thank you for stating in your Funding Statement:"This work was partially supported by the FHfFC (Future Home for Future Communications) project funded by Regione Lombardia. DB was the recipient.

GA : FHfFC 2016"

Reviewers' comments:

Reviewer's Responses to Questions

**Comments to the Author**

1. Is the manuscript technically sound, and do the data support the conclusions?

Reviewer #1: No

Reviewer #2: Yes

2. Has the statistical analysis been performed appropriately and rigorously? 

Reviewer #1: No

Reviewer #2: N/A

3. Have the authors made all data underlying the findings in their manuscript fully available?

Reviewer #1: No

Reviewer #2: Yes

4. Is the manuscript presented in an intelligible fashion and written in standard English?

Reviewer #1: Yes

Reviewer #2: Yes

5. Review Comments to the Author

Reviewer #1: The manuscript by Gianì et al. entitled “aTBP: a versatile tool for fish genotyping”, the author developed new genotyping method and applied to the identification of fish species. This article develops the new method to indicate the relationship of inter- and intra- species. But, I found there are some mistakes as well as experimental method in this manuscript. The author only uses seven fish species for this research and these samples come from the market. Farmed individuals may come from a single ancestor. The sample is too small and not representative. The author must use more species and evidence to prove that this method is useful. Teleost fishes represent a highly diverse group consisting of more than 20,000 species. The author cannot prove that this method can distinguish all species. The author must substantially modify the method and the description of the results. I consider the methodology of this article to be inappropriate in their current form, in my opinion, this manuscript does not meet criteria for publication and must therefore be reject.

Reviewer #2: The authors tried to prove that the popular aTBP method is a versatile tool for fish genotyping. In general, they provided solid data to support the main conclusions. However, minor revisions are required before acceptance for publication.

1. Extra editing is necessary.

2. The authors should discuss about how to deal with polyploid species, which would bring many more variants to improve the practical difficulty in genotyping. By the way, it would be much better if a practical database can be established. These issues should be mentioned in the discussion section.

6. PLOS authors have the option to publish the peer review history of their article (what does this mean?). If published, this will include your full peer review and any attached files.

Reviewer #1: No

Reviewer #2: No

---

## [Author Response · Author response to Decision Letter 0]

18 May 2020

REF 1

Q : The manuscript by Gianì et al. entitled “aTBP: a versatile tool for fish genotyping”, the author developed new genotyping method and applied to the identification of fish species. This article develops the new method to indicate the relationship of inter- and intra- species. But, I found there are some mistakes as well as experimental method in this manuscript. The author only uses seven fish species for this research and these samples come from the market. Farmed individuals may come from a single ancestor. The sample is too small and not representative. The author must use more species and evidence to prove that this method is useful. Teleost fishes represent a highly diverse group consisting of more than 20,000 species. The author cannot prove that this method can distinguish all species. The author must substantially modify the method and the description of the results. I consider the methodology of this article to be inappropriate in their current form, in my opinion, this manuscript does not meet criteria for publication and must therefore be reject.

A: We feel sorry to realise that our contribution has not found the appreciation of referee n.1. We are afraid that he/she could have possibly overlooked or missed some of the delivered information. With reference to the sampling, Ref.1 seems to have failed to appreciate that the vast majority of the analyzed fish samples came from aquaculture and were preliminary characterized at both morphological and at molecular levels with a panel of SSRs. We have clearly stated this in the M&M section and all over the manuscript, providing references to related projects and publications. This material was purposely and properly used to verify the reliability of our aTBP findings, acting as a gold standard. In addition we have collected samples from a completely different source, that is the fish market, authenticated with the COI marker so to build up a 3 markers crossed/referenced data : SSRs, COI and aTBP, an experimental strategy that serves the purpose of validating the aTBP method. The number of the analyzed species , seven, was limited by the availability of enough individuals and data that could corroborate the 3-markers approach and yet major species of commercial and scientific interests (tuna, sturgeon, gilthead seabream, European seabass, salmonids) have been included. Inclusion of additional fish species would have been redundant not adding more information, and would have made the paper too long and burdensome. With reference to the number of species distinguishable by aTBP, we certainly cannot exclude that out of 20.000 teleost species some aTBP profiles could overlap but that, in our opinion, would also be informative about their close genetic relationships very much similar to what we have shown for S. carpio, postulated as emerging from S. trutta fario as a recent speciation event. The common ancestor argument raised, is very much theoretical and in principle we cannot drop it but the experimental evidences we have collected from two different groups of sampling, fish market and aquaculture, where some intraspecific DNA polymorphisms are shared, tend to exclude it. Anyhow the aTBP tool applied to fish genotyping is also addressing the future, that is the monitoring of population changes that are in progress or will be in response to several adaptation events, thus providing a method that is very convenient , sustainable and affordable for many laboratories. 

 REF 2

Q: Extra editing is necessary.

A : It has been done in accordance to PLOS ONE style

Q. The authors should discuss about how to deal with polyploid species, which would bring many more variants to improve the practical difficulty in genotyping. By the way, it would be much better if a practical database can be established. These issues should be mentioned in the discussion section.

A: We thank the referee for his/her kind consideration of our work. With reference to ploidy, as it has been documented for sturgeon and the salmonids, known polyploid species with a prevalent 4N and 8N evolutionary/natural scale ploidy (4N functional scale) and very different chromosome numbers (from 54/58 to 240/264), aTBP is well performing and we think that its discrimination capacity could extend even further to a higher ploidy level since the major restriction of the technique is the limit of-resolution of the amplified fragments in CE, that is 1-2 bp. Clearly a higher number of target sequences would increase the number of peaks, not linearly because it would depend from the allelic variance, and this call for a faster reading, recognition and comparison of the profiles. This is the reason we are currently developing a software that could efficiently y compare the aTBP profile of the analyzed samples with authentic profiles of reference. This also calls for the establishment of a dedicated data base, as correctly suggested by the referee. We are doing this within the limit of our possibility because that requires a full collaboration with Institutions that can provide certified material . We are also in contact with some producers (caviar/ sturgeon) to expand our data. As requested by the referee we report these considerations in the discussion section together with a couple of new references.

---

## [Decision Letter · Decision Letter 1]

15 Jun 2020

PONE-D-20-08308R1

aTBP: a versatile tool for fish genotyping

PLOS ONE

Dear Dr. Breviario,

Thank you for submitting your manuscript to PLOS ONE. After careful consideration, we feel that it has merit but does not fully meet PLOS ONE’s publication criteria as it currently stands. Therefore, we invite you to submit a revised version of the manuscript that addresses the points raised during the review process.

We look forward to receiving your revised manuscript.

Kind regards,

Tzen-Yuh Chiang

Academic Editor

PLOS ONE

Reviewers' comments:

Reviewer's Responses to Questions

**Comments to the Author**

1. If the authors have adequately addressed your comments raised in a previous round of review and you feel that this manuscript is now acceptable for publication, you may indicate that here to bypass the “Comments to the Author” section, enter your conflict of interest statement in the “Confidential to Editor” section, and submit your "Accept" recommendation.

Reviewer #1: (No Response)

Reviewer #2: (No Response)

2. Is the manuscript technically sound, and do the data support the conclusions?

Reviewer #1: No

Reviewer #2: Yes

3. Has the statistical analysis been performed appropriately and rigorously? 

Reviewer #1: No

Reviewer #2: Yes

4. Have the authors made all data underlying the findings in their manuscript fully available?

Reviewer #1: No

Reviewer #2: Yes

5. Is the manuscript presented in an intelligible fashion and written in standard English?

Reviewer #1: Yes

Reviewer #2: No

6. Review Comments to the Author

Reviewer #1: In this study, the authors initially found size polymorphism in aTBP fragments, and examined this scenario further with evidence in other 7 fish species. The authors concluded the statement below, by the discovery of varying size fragments existing within and between species as well as at inter-population level,

“These data are discussed with respect to the application of the aTBP method to diverse fields of investigation that may include the characterization of a fish population and assessment of its variations in response to environmental changes, the recognition of genetic diversification resulting from hybridization events and studies on parental assignment as well as species traceability, authentication and detection in seafood.”

The above derivation, regarding to the function of aTBP, seems to be a courageous assumption while many unknows are yet, or left to be answered.

However, aTBP is a gene locus residing on the certain region of microsatellite, and the scoped and resolution in population genetic study would be constrained by analyzing various fragment sizes of homologous alleles of single gene, especially limited information revealed by fragment sizes only. Simply speaking, it is rarely, or nearly practical to study population genetic on targeted species through single genetic locus of one single microsatellite gene as found in this study.

Otherwise, fragment size is not recommended studying population genetic structure for its unknown mutation mode or mechanism affecting fragment length difference. For example, a set of homologous genes with similar genetic length may probably have difference nucleotide composition, in which a wrong conclusion might be easily obtained under this assumption.

By thoroughly consideration, despite the significant efforts from the authors, this paper of describing genotyping methodology to fish may present limited value and depth, as not claimed by the authors. Regretfully, I would suggest rejecting this paper as my final decision.

Reviewer #2: As mentioned in my previous comments, the authors should pay much attention to the overall writing of the manuscript. Extra editing from a professional company or a native English speaker is necessary. Other issues are Ok with good answers.

7. PLOS authors have the option to publish the peer review history of their article (what does this mean?). If published, this will include your full peer review and any attached files.

Reviewer #1: No

Reviewer #2: No

---

## [Author Response · Author response to Decision Letter 1]

24 Jun 2020

REF1 Remarks : This time ref1 objection, different from those made on the first version of the manuscript, seems to be motivated by his/her reluctancy to accept that aTBP can be a useful tool for studying popolulation genetics that is allele variance that may occur in a given popolutation across time and in response to different external changes. We say it seems because his/her consideration, as far as we can tell from his/her wording, starts from an ill-based assumption that is that aTBP is .. a gene locus residing on the certain region of microsatellite (cited) .. . As reported TBP is instead based on intron-length variation occurring in the numerous and different members of the beta-tubulin gene family. Thus, it is not a single locus marker neither a microsatellite sequence, not even … one single microsatellite gene (cited) . We are afraid that this misconception, although limited to a possible application of otherwise unquestioned experimental data, can lead to erroneous conclusions. Nevertheless, since it looks that ref 1 didn’t like the sentence of the abstract referring to the different applications of aTBP, with specific focus on population genetics , we have changed it to please him/her and make it even more fitting to the experimental data. 

REF2 remarks : With respect to ref 2 criticism about poor attention to the overall writing of the manuscript …and his/her recommendation for English editing .., please note that the text have been revised by Prof. Khidir Hilu of Virginia Tech, USA and Prof. Sara Patterson, Emeritus at Wisconsin University USA.

---

## [Decision Letter · Decision Letter 2]

21 Jul 2020

aTBP: a versatile tool for fish genotyping

PONE-D-20-08308R2

Dear Dr. Breviario,

We’re pleased to inform you that your manuscript has been judged scientifically suitable for publication and will be formally accepted for publication once it meets all outstanding technical requirements.

Kind regards,

Tzen-Yuh Chiang

Academic Editor

PLOS ONE

Additional Editor Comments (optional):

Reviewers' comments:

Reviewer's Responses to Questions

**Comments to the Author**

1. If the authors have adequately addressed your comments raised in a previous round of review and you feel that this manuscript is now acceptable for publication, you may indicate that here to bypass the “Comments to the Author” section, enter your conflict of interest statement in the “Confidential to Editor” section, and submit your "Accept" recommendation.

Reviewer #1: All comments have been addressed

Reviewer #2: All comments have been addressed

2. Is the manuscript technically sound, and do the data support the conclusions?

Reviewer #1: Yes

Reviewer #2: Yes

3. Has the statistical analysis been performed appropriately and rigorously? 

Reviewer #1: Yes

Reviewer #2: N/A

4. Have the authors made all data underlying the findings in their manuscript fully available?

Reviewer #1: Yes

Reviewer #2: Yes

5. Is the manuscript presented in an intelligible fashion and written in standard English?

Reviewer #1: Yes

Reviewer #2: Yes

6. Review Comments to the Author

Reviewer #1: After reviewing the manuscript titled “aTBP: a versatile tool for fish genotyping”, I feel that the manuscript has been significantly improved and satisfied with previous revisions. The data and analysis generally appear to be sound, the results are clear and interesting. In my opinion, this manuscript does meet criteria, and thus I feel should be published.

Reviewer #2: (No Response)

7. PLOS authors have the option to publish the peer review history of their article (what does this mean?). If published, this will include your full peer review and any attached files.

Reviewer #1: No

Reviewer #2: No

---

## [Editor Report · Acceptance letter]

23 Jul 2020

PONE-D-20-08308R2 

aTBP: a versatile tool for fish genotyping 

Dear Dr. Breviario:

I'm pleased to inform you that your manuscript has been deemed suitable for publication in PLOS ONE. Congratulations! Your manuscript is now with our production department. 

Kind regards, 

on behalf of

Dr. Tzen-Yuh Chiang 

Academic Editor

PLOS ONE